# MRSA Point Prevalence among Health Care Workers in German Rehabilitation Centers: A Multi-Center, Cross-Sectional Study in a Non-Outbreak Setting

**DOI:** 10.3390/ijerph16091660

**Published:** 2019-05-13

**Authors:** Melanie Schubert, Daniel Kämpf, Marlena Wahl, Samuel Hofmann, Maria Girbig, Lutz Jatzwauk, Claudia Peters, Albert Nienhaus, Andreas Seidler

**Affiliations:** 1Institute and Policlinic of Occupational and Social Medicine, Medical Faculty Carl Gustav Carus, Technische Universität Dresden, 01307 Dresden, Germany; daniel.kaempf@tu-dresden.de (D.K.); marlenawahl@yahoo.com (M.W.); samuel.hofmann@tu-dresden.de (S.H.); maria.girbig@tu-dresden.de (M.G.); andreas.seidler@mailbox.tu-dresden.de (A.S.); 2Department of Hospital Infection Control, Medical Faculty Carl Gustav Carus, Technische Universität Dresden, 01307 Dresden, Germany; lutz.jatzwauk@uniklinikum-dresden.de; 3Competence Centre for Epidemiology and Health Services Research for Healthcare Professionals (CVcare). University Medical Centre Hamburg-Eppendorf (UKE), 20251 Hamburg, Germany; c.peters@uke.de (C.P.); albert.nienhaus@bgw-online.de (A.N.); 4Department of Occupational Medicine, Public health and Hazardous Substances, Institution for Statutory Accident Insurance and Prevention in the Health and Welfare Services, 22089 Hamburg, Germany

**Keywords:** health care workers, MRSA prevalence, rehabilitation centers, non-outbreak, risk factors

## Abstract

People working in health care services have an increased risk of being infected with methicillin-resistant *Staphylococcus aureus* (MRSA), though little is known about the prevalence in rehabilitation centers. This cross-sectional study investigated the MRSA prevalence in employees from different rehabilitation centers and aimed to identify risk factors for MRSA transmission. We invited all staff (i.e., with and without patient contact from 22 participating rehabilitation centers; *n* = 2499) to participate. Study participation included a questionnaire on personal characteristics, lifestyle, personal and occupational risk factors for MRSA and nasal swabs taken by the study team. In total, 1005 persons participated in the study (response: 40.2%). Only four participants carried MRSA (0.40 (95% CI 0.00–1.00) per 100). MRSA carriage did not seem to be occupationally related, as it was found in different occupations with and without direct contact with MRSA patients, as well as in different clinics with different indications and patient clientele. We could not find a clear association between MRSA carriage and potential risk factors due to the low number of cases found. Genotyping revealed the *spa* types t032 (Barnim epidemic strain) and t1223. Our results suggest a low point prevalence of nasal MRSA colonization in a non-outbreak setting in employees from rehabilitation centers.

## 1. Introduction

Multi-resistant bacteria, such as the methicillin-resistant *Staphylococcus aureus* (MRSA), are important causes of nosocomial infections worldwide [1]. Whereas sufficient data for MRSA prevalence exist for acute care hospitals, this topic has been investigated only sparsely in rehabilitation centers. Rehabilitation centers usually provide care for patients that require further treatment to recover physiological or neurological functions, or to learn how to cope with residual impairments. Rehabilitation usually follows a holistic approach and is characterized by individuality, complexity, and multidisciplinarity. Thus, strict isolation of patients with MRSA (as in acute care hospitals) is usually not feasible for successful rehabilitation. For this reason, patients with MRSA might be refused by rehabilitation centers, and thus receive their rehabilitation late or not at all [2]. The handling of patients with MRSA has been subject to many debates, and recommendations have been proposed by several institutions [3,4,5,6]. In sum, an individual medical risk analysis with the aim to establish an optimal compromise between measures for the prevention of MRSA transmission and rehabilitation participation is recommended. Great emphasize is put on hygiene measures for staff and the patients. However, it has been shown that additional hygienic measures carry an additional substantial financial burden for rehabilitation centers, thereby lowering resources for rehabilitative core services [7]. 

Rehabilitation centers are often characterized by higher staff-patient contacts due to therapeutic measures. So far, there is only little data available for MRSA prevalence in health care workers (HCWs) in rehabilitation centers. A pooled analysis of studies from non-outbreak settings in Europe and the US showed a prevalence of MRSA carriage in HCWs of 1.8% (95% CI 1.34–2.50) [8]. In particular, Heudorf and colleagues observed MRSA prevalence of 0.6% in staff members from a German nursing home and geriatric rehabilitation unit. An Italian point prevalence study on MRSA colonization in wards with a mix of neurological, orthopedic, post-coma, and spinal cord injury cases found a prevalence of 3.1% [9]; this study observed MRSA colonization only among nurses but not among other HCWs. Moreover, a French study observed a MRSA prevalence of 10% among HWC staff and 20% among patients [10]. The highest MRSA prevalence was observed in assistant nurses, and occupations with more physical contact with patients were associated with a higher MRSA prevalence. The aim of this study was to determine the prevalence of MRSA in employees of different rehabilitation centers in a large multi-center setting in Germany and to identify risk factors for MRSA colonization.

## 2. Materials and Methods 

The study was conducted from September 2016 to May 2018. Ethical approval was obtained from the ethics committee of the TU Dresden (Clearance number EK 247062016). Methods were tested a priori with a pilot study in an acute care hospital in Dresden [11]. 

First, 34 rehabilitation centers in Saxony were invited, of which 10 participated in the study. These 10 rehabilitation centers employed 1173 persons altogether, of which 447 (38.1%) participated in the study. Twenty-four rehabilitation centers in Saxony refused to participate. Reasons for not participating, if named, were as follows: basic refusal to participate in studies of any kind (*n* = 1), not desired by the clinic management without specification (*n* = 3), fear regarding how to handle colonized staff and fear that a large number of employees would be on sick leave (*n* = 1), no one would participate (*n* = 1), expected high time expenditure (*n* = 1), no interest (*n* = 1), and no time (*n* = 1). Second, 66 rehabilitation centers from neighboring federal states, (i.e., Bavaria, Brandenburg, Saxony-Anhalt, and Thuringia) were also invited. Of these, 11 institutions participated in the study. One rehabilitation center contacted us through its own initiative.

In total, 22 German rehabilitation centers from different fields participated in the study. Specializations included addictions, cardiology, dermatology, gastroenterology, geriatrics, internal medicine, lymphology, mental trauma disorders, metabolic diseases, musculoskeletal disorders, neurology, orthopedics, oncology, pneumology, psychosomatics, psychotherapy, psychiatry, and rheumatic diseases. The number of employees within the rehabilitation centers ranged from 65–240. 

The study process began with an anonymous invitation that was distributed to all employees of a participating rehabilitation center. The invitation included information about the study, an informed consent form, and a questionnaire. Standard questions on personal characteristics (e.g., gender, age, etc.) and work, as well as questions concerning occupational and personal risk factors for MRSA were incorporated into the questionnaire. Questions concerning MRSA risk factors were predominately derived from a review by Albrich and Harbarth [12], a questionnaire used for staff in nursing homes by the German Social Accident Insurance Institution for the Health and Welfare Services (Berufsgenossenschaft für Gesundheitsdienst und Wohlfahrtspflege―BGW) [13], and additional literature. Furthermore, an on-site information event performed by the study team was offered to each rehabilitation center. 

For sampling of nasal swabs, the study team was on-site at the rehabilitation centers for 1–2 days. Participants were invited to an examination room, which was provided by each clinic. Before any measurements were taken, the informed consent and the completed questionnaire were collected and participants had an opportunity for asking questions. Then, samples were taken by a trained member of the study team using a swab from the anterior nares, which have been shown to be the main reservoir for MRSA [8]. Participants had the choice to either participate anonymously or to receive feedback of their MRSA status by mail to their private address. 

Prior to the study, training seasons for sampling of nasal swabs were conducted. Samples were usually taken by the study physician (DK) or prospective physicians (MW, SH). Sampling was done in both nostrils using sterile transystem^®^ cotton swabs (Hain LIFEscience GmbH, Nehren, Germany). One swab was used for both nostrils. Swab samples were analyzed by the Ostsächsische Labor according to the quality guidelines of the lab. The lab is accredited according to DAkkS: DIN EN ISO 15189. Prior to investigation of MRSA presence, broth enrichment media and MRSA selective agar plus non-selective blood agar were used in parallel. Swabs were plated on chromogenic agar plates selective for MRSA (BD BBL CHROMagar) and non-selective blood agar (Trypticase soy agar II with 5% sheep blood). In the case of no growth on the selective or blood agar, the enrichment broth was plated on the selective agar to culture low MRSA cell numbers. Suspicious colonies underwent phenotypical confirmation using an immunochromatic qualitative PBP2a assay (Alere) as well as antimicrobial susceptibility testing to measure MICs of oxacillin and cefoxitin using broth microdilution tests. In most cases, one single isolate was tested for MRSA unless colonies with different morphologies were present either on the selective or blood agar plate. Only *S. aureus* isolates suspicious for MRSA were further tested in the study and data concerning *spa* types of MSSA can therefore not be presented. Suspect MRSA isolates were definitively identified by the German National Reference Center for Staphylococci and Enterococci (Nationales Referenzzentrum für Staphylokokken und Enterokokken) in Wernigerode, Germany. The presence of mecA was detected in all samples. 

Descriptive statistics are shown for all data. Odds ratios and 95% confidence intervals were calculated using the exact method in Stata (StataCorp. 2011. Stata Statistical Software: Release 12. StataCorp LP, College Station, TX, USA).

Overall, 2499 employees were invited, of which 1005 participated in the study. Thus, the response rate was 40.2%. Within rehabilitation centers there was a huge variation in response rates, ranging from 15.2% to 75.0%. Most participants received the feedback by mail (*n* = 988); only 17 persons participated anonymously in the study.

## 3. Results

### 3.1. Charcterization of the Study Population

Mainly women participated in the study (82%, Table 1). Most participants were between 50 and 59 years old (31.2%) and were secondary school graduates (49.6%). Participants predominately lived in a partnership (78.8%). Mostly therapists (33.9%) participated in the study with a professional experience of 15–30 years (34.1%). Only about one-third (32.3%) reported having had close patient contact within the last four weeks. 

Having contact with MRSA-patients within the last four weeks was reported by 13.5%, of which the majority (76.5%) also reported wearing protective gear at all times; 22.1% of participants had occasional contact without protective clothing, and two participants did not answer the question. When having contact with MRSA patients, the majority of participants (58.1%) reported wearing a surgical face mask, disposable gloves, and a lab coat.

In terms of private life risks, only a few participants had known contact with MRSA carriers (1.4%). Approximately 7.2% were caring for relatives at home and 13.6% had contact with persons in need of care within the last 4 weeks. Only 16 of 1005 reported working in the ambulant sector besides their work at the rehabilitation center. The majority of participants had contact with pets (61.2%). Contact with farm animals was reported by 8.9%.

Seventeen of 1005 participants had previously had MRSA in the past. About one-third (30.2%) of participants used antibiotics within the last 12 months. A hospital stay within the last 12 months was reported by 9.5%, and 4.0% had an operation within the last 30 days. About 10% reported suffering from chronic skin disease or chronic respiratory disease. Characteristics of study population are displayed in Table 1.

### 3.2. Prevalence of MRSA and Risk Factors

Four of 1005 participants tested positive as MRSA carriers during the examination, corresponding to a prevalence of 0.40% (95% CI 0.00–1.00). MRSA carriage was found in four different clinics with the following specializations: geriatrics, neurology, psychosomatics, and cardiology/oncology/gastroenterology. The MRSA carriers were working as a physician (*n* = 1) or as a cleaning staff member (*n* = 1), and the two remaining MRSA carriers stated having a job not listed in the questionnaire (i.e., geriatrics and psychosomatics, without any further specification). Professional experience ranged between MRSA carriers from less than 1 year (*n* = 1), 1–5 years (*n* = 1), 6–10 years (*n* = 1), and 15–30 years (*n* = 1). Three MRSA carriers were women and one was a man. 

Two of the MRSA carriers had close contact with patients and MRSA patients within the last 4 weeks. 

The contact with MRSA patients was always with protective work clothing (surgical face mask, disposable gloves, and lab coat). Two of the MRSA carriers did not know if they had contact with patients with MRSA. 

One MRSA carrier reported working in the Czech Republic within the last 12 months. Regarding private-life factors, none of the four MRSA carriers were caring for relatives at home, had contact with MRSA carriers or with persons in need of care within the last 4 weeks, or worked in the ambulant sector outside work. Only one of the four MRSA carriers had contact with pets and none had contact with farm animals. For risk factors associated with health status, the following picture emerged: none of the MRSA carriers had a chronic skin or respiratory disease, diabetes mellitus, used antibiotics, or had an inpatient hospital stay within the last 12 months. One MRSA carrier had an operation within the last 30 days. Another carrier had a history of MRSA colonization. 

Altogether, we could not find a clear association between MRSA carriage and potential risk factors due to the low sample size. We could not clearly identify risk factors for MRSA carriage, but the results did not contradict previous risk factor research. Having contact with patients with MRSA within the last four weeks, working abroad, previous MRSA carriage, and an operation within the last 30 days may increase the risk of MRSA carriage. Results are presented in Table 2. 

### 3.3. MRSA Genotyping

Genotyping revealed the MRSA *spa* types t032 (CC22, mecA positive) and t223 (CC22, mecA positive). The *spa* type t032 occurred in a clinic for geriatrics and a clinic for cardiology/oncology/ gastroenterology. The tetracycline-resistant MRSA *spa* type t223 was observed in a psychosomatic rehab center with psychotherapy. Unfortunately, we were unable to genotype the fourth MRSA-positive sample. 

## 4. Discussion

The present study investigated the prevalence of MRSA in 22 rehabilitation centers with different specializations. Only four of 1005 participants were colonized with MRSA (0.4%, 95% CI 0.0–1.0%). This corresponds to results of the German nursing home and geriatric rehabilitation unit, which showed a prevalence of MRSA in HCWs of 0.6% [14]. For HCWs working in German acute care hospitals, prevalence rates of 2.3–5.3% have been described [15,16,17,18]. This suggests that MRSA prevalence might be generally lower in rehabilitation centers than in acute care clinics. Nevertheless, we also recently studied MRSA prevalence in an acute care hospital in a small pilot study. Here only 1 in 180 (0.6%) employees, or 1 of 149 (0.7%) persons with close patient contact, carried MRSA [11]. A general decrease in MRSA has been observed for Germany and Europe [19,20]. Thus, the low observed MRSA prevalence may also reflect this decrease. 

MRSA colonization was observed in four different rehabilitation centers with the following different specialties: geriatrics, neurology, psychosomatics, and a mix of cardiac/oncological/gastroenterological cases. Geriatrics and neurology have been described as high risk units for MRSA for patients. In contrast, psychosomatic clinics and units with cardiac/oncological/gastroenterological cases are considered low risk MRSA areas [21,22]. One MRSA carrier was a physician, one worked as a cleaner, and two stated as having a job not listed in the questionnaire. Only two of the four MRSA carriers had close contact with patients and MRSA patients within the last 4 weeks (neurology and geriatrics). Thus, it is possible that MRSA transmission may have occurred indirectly. MRSA strains have a high tenacity and die off slowly (i.e., they can survive for longer than 6 months) [23]. MRSA can be found on general surfaces such as beds, floor, linen, curtains, and overbed tables [24]. Boyce and colleagues showed that environmental contamination with MRSA occurred in 69% of colonized patients [25]. Moreover, it has been shown that 42% of persons not having direct contact with MRSA patients but touching contaminated surfaces had MRSA on the gloves. This may suggest that MRSA transmission might have occurred from contaminated surfaces without having direct patient contact in two cases. Thus, strict hygiene measures, including routine disinfection of contact surfaces, are essential for limiting MRSA transmission [3,24]. However, since we do not have data on patient clientele and patient MRSA status, we cannot make an assumption about transmission pathways. Moreover, MRSA transmission may have also occurred outside the centers.

When investigating risk factors for MRSA carriage, we could not find a clear association between MRSA carriage and potential personal and occupational risk factors due to the low number of MRSA carriers in the study sample―only four HCWs carried MRSA. Nevertheless, the results are not contradictory to previous research on MRSA risk factors. Having contact with patients with MRSA within the last four weeks, working abroad, known history of MRSA carriage, and having had an operation within the last 30 days may increase the risk of MRSA carriage. These are well known risk factors for MRSA transmission in HCWs [15,26]. 

In this study, two different MRSA *spa* types of the epidemic strain CC22 (mecA positive) were identified: t032 and t223. The Barnim epidemic strain (t023) was observed in the rehabilitation centers with geriatric patients and a mix of cardiological/oncological/gastroenterological cases. This strain is a classical nosocomial MRSA strain that is spread throughout Germany [20,27]. Rehabilitation centers usually provide care following treatment in an acute care hospital. Thus, the spread of MRSA between hospitals and rehabilitation centers is likely to occur. Indeed, the Barnim epidemic strain has been shown to be the second most frequent epidemic strain in patients from rehabilitation centers with neurologic, orthopedics, internal medicine, and geriatric units [28]. The *spa* type t223 occurred in a psychosomatic rehabilitation center. This strain is associated with community-acquired MRSA and has only been observed sporadically before 2014 [20]. A recent analysis by the Robert Koch Institute showed a trend for increasing tetra-cycline resistant MRSA in recent years, where it has been observed among asylum seekers in central reception centers [20]. 

One study limitation was the low response rate of 40.2%. The variation in response was rather large, ranging from 15.2% to 75.0% within the single rehabilitation centers. Mainly women participated in the study (82%). Most participants were between 50–59 years (31.2%) followed by 40–49 years (24.2%), 30–39 years (22.8%), under 30 years (10.9%), and over 60 years (8.8%). Unfortunately, we do not have information about the general employee characteristics of the rehabilitation centers. However, according to the German Federal Statistical Office (DESTATIS), about 77% of health personnel in prevention and rehabilitation centers were women in 2016 [29]. Moreover, the age structure of study participants generally corresponds to the age structure of rehabilitation staff in Germany, as monitored by DESTATIS. In the year 2016, the majority of staff working in prevention and rehabilitation centers were between 50–60 years (31.4%) followed by 40–50 years (24.0%), 30–40 years (19.0%), under 30 years (14.9%), and over 60 years (10.7%). Thus, the characteristics of our study population generally correspond to that of nationwide rehabilitation staff characteristics, suggesting little bias. Another limitation of the study is the lack of environmental samples in the participating centers. A further study limitation is that we only tested employees, and not patients. Data on patient MRSA status would have made it possible to test assumptions about possible transmission paths [30]. Moreover, we only sampled the anterior nares for MRSA. This may have led to an underestimation of MRSA prevalence because it has been shown that screening other body sites (e.g., throat and axilla) increases MRSA yield over nares alone [31]. The study was also underpowered due to the (unexpectedly) low sample of MRSA carriers. Due to the low prevalence of MRSA carriage in this population, a huge sample size is required to generate robust data for identifying risk factors for MRSA transmission. 

## 5. Conclusions

Our results suggest a low point prevalence of MRSA in employees from German rehabilitation centers in a non-outbreak setting. MRSA carriage was a rare event found in different occupations among workers with and without direct contact with MRSA patients, as well as in different clinics with different specializations and caring for different patient clientele. Due to the low prevalence of MRSA, risk factors for MRSA carriage could not be evaluated. Altogether, the results correspond to the recently reported trend of decreasing MRSA prevalence in Europe. 

## Figures and Tables

**Table 1 ijerph-16-01660-t001:** Participant characteristics and methicillin-resistant *Staphylococcus aureus* (MRSA) risk factors.

Characteristics	Number	Percent
Total	1005	100
Sex		
Female	824	82.0
Male	175	17.4
Not reported	6	0.6
Age in years		
<20	6	0.6
20–29	104	10.3
30–39	229	22.8
40–49	243	24.2
50–59	314	31.2
≥60	88	8.8
Not reported	21	2.1
Education		
Secondary school graduation	498	49.6
High school, university entrance qualification	440	43.8
Other	56	5.6
No degree	1	0.1
Not reported	10	1.0
Occupation		
Physician	104	10.3
Nurse	140	13.9
Therapist	338	33.6
Medical technical assistant	28	2.8
Transport service workers	3	0.3
Cleaner	49	4.9
Administrative personnel	148	14.7
Others	186	18.5
Not reported	9	0.9
Professional experience in years		
≤1	73	7.3
1–5	232	23.1
6–10	145	14.4
11–20	111	11.0
21–40	343	34.1
>41	95	9.5
Not reported	6	0.6
Partnership		
Yes	792	78.8
No	200	19.8
Not reported	13	1.3
Household Size		
One-person	199	19.8
Multi-person	792	78.8
Not reported	14	1.4
**Work-related MRSA risk factors**		
Close contact with patients within the last 4 weeks (washing, dressing, changing bandages, etc.)		
Yes	325	66.3
No	666	32.3
Not reported	14	1.4
Contact with MRSA patients within the last 4 weeks		
Yes	136	13.5
No	320	31.8
Unknown	542	53.9
Not reported	7	0.7
Wearing protective gear when having contact with MRSA patients		
Yes, always	104	76.5
Occasionally without	30	22.1
Not reported	2	1.5
Protective gear worn when having contact with MRSA patients		
Surgical face mask, disposable gloves, and lab coat	79	58.1
Surgical face mask and disposable gloves	12	8.8
Disposable gloves	11	8.1
Disposable gloves and lab coat	2	1.5
Surgical face mask and lab coat	1	0.7
Not applicable	2	1.5
None reported	29	21.3
Working abroad		
Yes	15	1.5
No	978	97.3
Not reported	12	1.2
**Private life-related MRSA risk factors**		
Caring for relatives at home		
Yes	72	7.2
No	926	92.1
Not reported	7	0.7
Contact with MRSA carriers		
Yes	14	1.4
No	535	53.2
Unknown	449	44.7
Not reported	7	0.7
Contact with persons in need of care within the last 4 weeks		
Yes	137	13.6
No	863	85.9
Not reported	5	0.5
Working in the ambulant sector outside of work at rehabilitation center		
Yes	16	1.6
No	984	97.9
Not reported	5	0.5
Contact with farm animals		
Yes	89	8.9
No	910	90.5
Not reported	6	0.6
Contact with pets		
Yes	615	61.2
No	387	38.5
Not reported	3	0.3
**Risk factors associated with health status**		
Chronic skin disease		
Yes, own diagnosis	14	1.4
Yes, doctor’s diagnosis	97	9.7
No	876	87.2
Not reported	18	1.8
Chronic respiratory disease		
Yes, own diagnosis	25	2.5
Yes, doctor’s diagnosis	90	9.0
No	869	86.5
Not reported	21	2.1
Diabetes mellitus		
Yes, own diagnosis	2	0.2
Yes, doctor’s diagnosis	25	2.5
No	951	94.6
Not reported	27	2.7
Previous MRSA carriage		
Yes	17	1.7
No	983	97.8
Not reported	5	0.5
Use of antibiotics within the last12 months		
Yes	304	30.2
No	690	68.7
Not reported	11	1.1
Having had an operation within the last 30 days		
Yes	40	4.0
No	959	95.4
Not reported	6	0.6
Inpatient hospital treatment within the last 12 months		
Yes	95	9.5
No	902	89.8
Not reported	8	0.8

**Table 2 ijerph-16-01660-t002:** Prevalence (odds ratios (ORs)) for MRSA risk factors.

Risk Factor	MRSA Positive (*n*)		MRSA Negative (*n*)		OR	95% CI
	No (Reference)	Yes	No (Reference)	Yes		
**Work-related risk factors**						
Close contact with patients within the last 4 weeks	2	2	664	323	2.1	0.1–28.5
Contact with MRSA patients within the last 4 weeks	2	2	860	134	6.4	0.5–89.0
Working abroad	3	1	975	14	23.2	0.4–305.5
**Private life-related MRSA risk factors**						
Caring for relatives at home	4	0	922	72	-	-
Contact with MRSA carriers	4	0	980	14	-	-
Contact with persons in need of care within the last 4 weeks	4	0	859	137	-	-
Working in the ambulant sector outside of work	4	0	980	16	-	-
Contact with farm animals	4	0	906	89	-	-
Contact with pets	3	1	384	614	0.2	0.0–2.6
**Risk factors associated with health status**						
Chronic skin disease	4	0	872	111	-	-
Chronic respiratory disease	4	0	865	115	-	-
Diabetes mellitus	4	0	947	27		
Previous MRSA carriage	3	1	980	16	20.4	0.4–266.8
Antibiotics use within the last12 months	3	0	687	304	-	-
Operation within the last 30 days	3	1	956	39	8.2	0.2–103.8
Inpatient hospital treatment within the last 12 months	4	0	898	95	-	-

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
