# Peer review of "MRSA Point Prevalence among Health Care Workers in German Rehabilitation Centers: A Multi-Center, Cross-Sectional Study in a Non-Outbreak Setting"

_ijerph, 2019, doi:10.3390/ijerph16091660_

Reviewer 1 Report

GENERAL COMMENTS: This study investigated the point prevalence of MRSA carriage among workers in 22 rehabilitation centers in Germany between September 2016 and May 2018 in non-outbreak settings. A total of 1,005 individual staff participated in the study. Nasal swabs were taken from the participants for MRSA recovery. Only 4/1005 participants were found to carry nasal MRSA. This MRSA prevalence rate of 0.4% is very low indeed relative to other studies of healthcare workers in nursing homes in other countries.

The study is large-scale and the findings are very significant if they can be believed. I have provided some specific comments below for the authors’ consideration. These are intended to be constructive and to improve clarity and impact.

SPECIFIC COMMENTS:

(1). No details whatsoever are provided concerning the methodology used to detect and recover MRSA from the nasal swab samples taken from the study participants. This leads me to suspect that the methodology may not have been sensitive or robust enough to detect MRSA or to detect MRSA from low-abundance carriers. This is a major weakness of the study. The precise methodology, including the type of swabs used has to be provided in the manuscript.

(2). Were all nasal swab samples taken by the same individual? If many individuals took the swab samples-this introduces a variable.

(3). Were swab samples subjected to broth enrichment in media containing high salt to select for staphylococci prior to investigating for  the presence of MRSA?

(4). Were swab samples/enrichment cultures plated on chromogenic media selective for (i) Staphylococcus aureus,(ii) MRSA or (iii) on media selective for staphylococci (e.g. mannitol salt agar).

(5). How were MRSA definitively identified (e.g. detection of mecA or mecC genes and/or cefoxitin resistance)?

(6). How many individual S. aureus colonies recovered from individual swab/enrichment cultures were tested for MRSA.

(7). I expect that the prevalence of methicillin-susceptible S. aureus (MSSA) recovered from the nasal swabs of the 1,005 participants included in the study would have been determined. Other studies have shown the rate of nasal MSSA carriage of between 20-35%. This data has to be included in the study because MSSA are often just as pathogenic as MRSA in terms of virulence determinants carried. Including this data would add significant depth and value to the present study. The range of spa types among the MSSA should also be presented. This will be diverse.

 Author Response

(1). No details whatsoever are provided concerning the methodology used to detect and recover MRSA from the nasal swab samples taken from the study participants. This leads me to suspect that the methodology may not have been sensitive or robust enough to detect MRSA or to detect MRSA from low-abundance carriers. This is a major weakness of the study. The precise methodology, including the type of swabs used has to be provided in the manuscript.

Thank you very much for your valuable opinion. We included information on methodology in the text (starting line 106)

(2). Were all nasal swab samples taken by the same individual? If many individuals took the swab samples-this introduces a variable.

Thanks, you make a valid point. Prior to the study, training seasons for sampling of nasal swabs were conducted. Samples were usually taken by the study physician (DK) or prospective physicians (MW, SH). Information was included in line 106-107.

(3). Were swab samples subjected to broth enrichment in media containing high salt to select for staphylococci prior to investigating for the presence of MRSA?

We used broth enrichment media and MRSA selective agar plus non-selective blood agar (TSA with sheep blood) in parallel. In case of no growth on the selective or blood agar, the enrichment broth was plated on the selective agar to culture low MRSA cell numbers. Information is included in line 110-112.

(4). Were swab samples/enrichment cultures plated on chromogenic media selective for (i) Staphylococcus aureus,(ii) MRSA or (iii) on media selective for staphylococci (e.g. mannitol salt agar).

Swabs were plated on Chromogenic agar plates selective for MRSA (BD BBL CHROMagar) and non-selective blood agar (Trypticase Soy Agar II with 5% Sheep Blood) (line 112-114).

(5). How were MRSA definitively identified (e.g. detection of mecA or mecC genes and/or cefoxitin resistance)?

Suspicious colonies underwent phenotypical confirmation using an immunochromatographic qualitative PBP2a assay (Alere) as well as antimicrobial susceptibility testing to measure MICs of oxacillin and cefoxitin using broth microdilution tests. All positive isolates were finally confirmed by the German national reference center for staphylococci and enterococci (Nationales Referenzzentrum für Staphylokokken und Enterokokken) in Wernigerode, Germany (line: 116-123).

(6). How many individual S. aureus colonies recovered from individual swab/enrichment cultures were tested for MRSA.

In most cases one single isolate was tested for MRSA unless colonies with different colony morphologies were present either on the selective or blood agar plate (line 117-119).

(7). I expect that the prevalence of methicillin-susceptible S. aureus (MSSA) recovered from the nasal swabs of the 1,005 participants included in the study would have been determined. Other studies have shown the rate of nasal MSSA carriage of between 20-35%. This data has to be included in the study because MSSA are often just as pathogenic as MRSA in terms of virulence determinants carried. Including this data would add significant depth and value to the present study. The range of spa types among the MSSA should also be presented. This will be diverse.

We agree that the determination of MSSA would have added interesting information. But please consider that the study was carried out in the context of occupational safety in order to facilitate the recognition of MRSA as an occupational disease in Germany. The determination of MSSA was not included in the budget. Only S.aureus isolates suspicious for MRSA were further tested in this study and therefore data concerning spa types of MSSA cannot be presented unfortunately.

Reviewer 2 Report

The authors presented an interesting study evaluating the prevalence of MRSA carriage among health care workers (HCWs) in different German Rehabilitation Centers. They found a very low prevalence, in line with national data in the literature.

The manuscript represents original work and it is well written.

I have only few comments:

- the authors must include in the title that the carriage of MRSA was explored in HCWs. As it is presented, it is not clear.

- the authors cannot evaluate risk factors for MRSA carriage due to the very low prevalence (only 4 HCWs carried MRSA). They should better underline it in the manuscript.

- line 159: remove possibly

- lines 159-161: the sentence is too strong: please rephrase it considering the main limitation of the study (very low prevalence) which makes it impossible to define risk factors for MRSA

- another limitation of the study is the lack of environmental samples in the participating centers. Please insert it in the discussion.

- lines 210-204: the sentence is too strong: please rephrase it considering the main limitation of the study (very low prevalence) which makes it impossible to define risk factors for MRSA, even suggestions.

- Table1. Insert in the column the total number of HCWs included in the study.

Author Response

Thank you really much for your rigorous review.

I have only few comments:

- the authors must include in the title that the carriage of MRSA was explored in HCWs. As it is presented, it is not clear.

Thank you very much for this remark. The title was changed to “MRSA point prevalence among health care workers in German rehabilitation centers: A multi-center, cross-sectional study in a non-outbreak setting.”

- the authors cannot evaluate risk factors for MRSA carriage due to the very low prevalence (only 4 HCWs carried MRSA). They should better underline it in the manuscript.

We have tried to better underline this fact in the manuscript (e.g. line 220, 262).

- line 159: remove possibly

“Possibly” was removed (line 176).

- lines 159-161: the sentence is too strong: please rephrase it considering the main limitation of the study (very low prevalence) which makes it impossible to define risk factors for MRSA

Thank you for this important remark. We rephrased the sentence to: “We could not clearly identify risk factors for MRSA carriage but the results did not contradict previous risk factor research.  Having contact to patients with MRSA within the last four weeks, working abroad, previous MRSA carriage and an operation within the last 30 days may increase the risk of MRSA carriage.” (line 176-179)

- another limitation of the study is the lack of environmental samples in the participating centers. Please insert it in the discussion.

Thank you, you make a valid point here. We included this limitation in line (line 247-248).

- lines 210-204: the sentence is too strong: please rephrase it considering the main limitation of the study (very low prevalence) which makes it impossible to define risk factors for MRSA, even suggestions.

We rephrased the sentence to: “Nevertheless, the results are not contradictory to previous research on MRSA risk factors. Having contact to patients with MRSA within the last four weeks, working abroad, known history of MRSA carriage and an operation within the last 30 days may increase the risk of MRSA carriage.” (line 220-223)

- Table1. Insert in the column the total number of HCWs included in the study.

We have added the total number to the table.

Reviewer 3 Report

Thank you for the opportunity to review your article.

Specific comments:

78; Add a space before "Second"

118-120: the authord should put these information on the women and age of participants in the Results section

157: change "one person" with "another carrier

177: add the percentage of 1 on 180 within brackets

178: add the percentage of 1 on 149 within brackets

Author Response

78; Add a space before "Second"

Thanks a lot for taking your precious time for reviewing our manuscript. We have corrected this (line 79).

118-120: the authord should put these information on the women and age of participants in the Results section:

The information is included in the results, in the section “Characterization of study population” (starting line 131).

157: change "one person" with "another carrier:

Thank you very much for your remark, we have changed “one person” to “another carrier” (line 174).

177: add the percentage of 1 on 180 within brackets:

We have included the percentage in brackets (line 196).

178: add the percentage of 1 on 149 within brackets:

We have included the percentage in brackets (line 196).

Reviewer 4 Report

This research article is addressing a very important topic, the prevalence or MRSA in rehabilitation centres, which has not been studied properly before. The writing style is excellent, the narrative argument is clear and it is very easy to follow the authors. There are no concerns in terms of ethics or the methodology used. The number of centres and participants enrolled in the study is impressive. Despite of the low number of MRSA isolates gathered, the results could be very informative, particularly when compared to the data published in other countries. One minor comment, you should write in italics the species names included in the reference list.

Author Response

Thanks a lot for taking your precious time for reviewing our manuscript and for your kind words. We have corrected the species name in italics.

Round  2

Reviewer 1 Report

The authors' have addressed most of my original comments satisfactorily. The revised manuscript is much stronger. I have one final minor comment that the authors' should consider:

The authors state on lines 122-123 of the revised manuscript that “All positive isolates were finally confirmed by the German national reference center forstaphylococci and enterococci (Nationales Referenzzentrum für Staphylokokken und Enterokokken)123 in Wernigerode, Germany. I suggest that this should be changed to:

“Suspect MRSA isolates were definitively identified by the German national reference center forstaphylococci and enterococci (Nationales Referenzzentrum für Staphylokokken und Enterokokken)123 in Wernigerode, Germany”.

Please clarify that the Reference Centre detected the presence of mecAor mecCas part of the identification of MRSA.

Author Response

Dear reviewer,

thank you very much for your rigorous review. The sentence was changed according to your suggestion (line 121-123): Suspect MRSA isolates were definitively identified by the German national reference center forstaphylococci and enterococci (Nationales Referenzzentrum für Staphylokokken und Enterokokken)123 in Wernigerode, Germany”.

Please clarify that the Reference Centre detected the presence of mecAor mecCas part of the identification of MRSA.

Thank you very much for this remark. The presence of mecA was detected. This information was included in line 123.

We hope we addressed your comments to your satisfaction.

Best wishes, Melanie Schubert